# The Protective Role of Interleukin-37 in Cardiovascular Diseases through Ferroptosis Modulation

**DOI:** 10.3390/ijms25189758

**Published:** 2024-09-10

**Authors:** Alfredo Cruz-Gregorio, Luis M. Amezcua-Guerra, Brandon Fisher-Bautista, Abraham Romero-Beltrán, Gabriela Fonseca-Camarillo

**Affiliations:** 1Departamento de Fisiología, Instituto Nacional de Cardiología Ignacio Chávez, Ciudad de México 14080, Mexico; alfredo.cruz@cardiologia.org.mx; 2Departamento de Inmunología, Instituto Nacional de Cardiología Ignacio Chávez, Ciudad de México 14080, Mexico; lmamezcuag@gmail.com (L.M.A.-G.); brandoncamazotz@gmail.com (B.F.-B.); abraham6romero@gmail.com (A.R.-B.); 3Departamento de Atención a la Salud, División de Ciencias Biológicas y de la Salud, Universidad Autónoma Metropolitana Unidad Xochimilco, Ciudad de México 14387, Mexico; 4Programa de Maestría en Ciencias Químico Biológicas, Instituto Politécnico Nacional, Ciudad de México 11350, Mexico

**Keywords:** interleukin-37, ferroptosis, macrophages, atherosclerosis, cardiovascular diseases, inflammation

## Abstract

The role of ferroptosis and iron metabolism dysregulation in the pathophysiology of cardiovascular diseases is increasingly recognized. Conditions such as hypertension, cardiomyopathy, atherosclerosis, myocardial ischemia/reperfusion injury, heart failure, and cardiovascular complications associated with COVID-19 have been linked to these processes. Inflammation is central to these conditions, prompting exploration into the inflammatory and immunoregulatory molecular pathways that mediate ferroptosis and its contribution to cardiovascular disease progression. Notably, emerging evidence highlights interleukin-37 as a protective cytokine with the ability to activate the nuclear factor erythroid 2-related factor 2 pathway, inhibit macrophage ferroptosis, and attenuate atherosclerosis progression in murine models. However, a comprehensive review focusing on interleukin-37 and its protective role against ferroptosis in CVD is currently lacking. This review aims to fill this gap by summarizing existing knowledge on interleukin-37, including its regulatory functions and impact on ferroptosis in conditions such as atherosclerosis and myocardial infarction. We also explore experimental strategies and propose that targeting interleukin-37 to modulate ferroptosis presents a promising therapeutic approach for the prevention and treatment of cardiovascular diseases.

## 1. Introduction

Ferroptosis is a novel form of modulated, non-apoptotic cell death, characterized by iron dysregulation, oxidative stress, and the accumulation of lipid peroxides. This process leads to damage of the cell membrane and subsequent cell death caused by oxidative injury [1,2,3,4,5]. The interplay between oxidative damage and cell death may trigger inflammation, which in turn can promote ferroptosis, establishing a self-amplifying positive feedback loop. Both ferroptosis and inflammation are emerging as promising therapeutic targets in several pathological conditions, including cardiovascular diseases (CVDs) [6].

Within the context of inflammation, macrophages play a fundamental role in the progression of CVDs, such as coronary artery disease, peripheral artery disease, and aortic atherosclerosis. Interleukin (IL)-37, a member of the IL-1 cytokine family, is recognized for its ability to downregulate pro-inflammatory cytokines and inhibit innate immune responses [7]. IL-37 is constitutively expressed in immune cells, including macrophages, as well as regulatory cells like B and T regulatory cells, with its expression being upregulated in response to pro-inflammatory stimuli [8,9]. Notably, IL-37 has been shown to reduce macrophage ferroptosis by upregulating the nuclear factor erythroid 2-related factor 2 (NRF2) pathway, thereby attenuating the progression of atherosclerosis in murine models [10].

Despite growing evidence supporting its role, there has been no comprehensive review of IL-37’s protective effect against ferroptosis in CVDs. In this review, we consolidate the current knowledge on IL-37 as a regulatory cytokine and its immunomodulatory role in ferroptosis within the context of CVDs. We discuss experimental approaches and provide insights into potential therapeutic strategies that target IL-37 via modulation of ferroptosis, emphasizing its potential as a novel strategy for the prevention and treatment of CVDs.

## 2. Ferroptosis and Cardiovascular Diseases

Excessive iron concentrations contribute to the production of reactive oxygen species (ROS) through iron-dependent Fenton and Haber–Weiss reactions. Although ROS production is regulated by enzymatic and non-enzymatic antioxidants, depletion of these antioxidants can lead to an overproduction of ROS, resulting in oxidative stress and oxidative damage. Under these conditions, lipoxygenase (LOX) enzymes and the overproduction of ROS generate lipid radicals during lipid peroxidation. Usually, glutathione peroxidase 4 (GPX4) degrades lipid radicals; however, when GPX4 is dysfunctional, lipid radicals accumulate, damaging biomolecules such as proteins, DNA, and lipids. Since cell membranes are composed of phospholipids, lipid peroxidation disrupts the cell membrane integrity and fluidity, leading to cell death associated with oxidative damage and ferroptosis (Figure 1) [1,2].

Although oxidative damage-related cell death has been studied for years, the term “ferroptosis”, describing a form of programmed cell death distinct from apoptosis, autophagy, or necroptosis, was introduced by Dixon et al. in 2012 [11]. This group discovered that erastin decreases cysteine uptake, thereby inhibiting glutathione (GSH) synthesis—an essential cofactor for GPX4 to reduce the accumulation of lipid peroxides. The reduction of GSH, coupled with increased iron levels, significantly increases ROS production and, consequently, lipid peroxides. Oxidative damage resulting from this process can be observed under microscopy as increased plasma bilayer membrane density, reduced mitochondrial size, loss or reduction of mitochondrial cristae, and mitochondrial condensation, swelling, and membrane rupture, all hallmarks of cell death due to ferroptosis [12].

Ferroptosis is distinct from other types of programmed cell death, such as apoptosis, as it does not involve caspase activation nor does it disrupt DNA structure. Instead, cells undergoing ferroptosis exhibit necrotic-like morphology, characterized by a lack of chromatin condensation, increased membrane density, and rupture of the outer cell membrane [13]. This rupture is related to the formation of membrane nanopores, which facilitate cell bursting [14]. As a result, ferroptosis differs in its morphology, biochemistry, and genetics from other forms of programmed cell death. Moreover, unlike apoptosis, ferroptosis is inhibited by iron-chelating agents or antioxidants, rather than caspase inhibitors [14]

Dixon et al. also demonstrated that ferroptosis, induced by agents as erastin or 3R-RSL3 (RSL3), increases intracellular iron levels [11]. During Fenton and Haber–Weiss reactions, iron directly contributes to excessive ROS generation, promoting lipid peroxidation. Additionally, iron overload is associated with the increased activity of LOX enzymes, which further promote lipid peroxidation [15]. Given the central role of iron in ferroptosis induction, different genes and proteins involved in iron homeostasis—including those regulating iron import, export, storage, and regeneration—can influence susceptibility to ferroptosis [16]. For instance, in the digestive system, gastric acid reduces Fe^3+^ to Fe^2+^, which is then absorbed in the duodenum and jejunum. Once inside the cell, Fe^2+^ is oxidized back to Fe^3+^ by the action of ceruloplasmin at the cell membrane, where it binds to transferrin (TF) to form the TF-Fe^3+^ complex. This complex interacts with transferrin receptor 1 to be endocytosed by the cell. Inside the cell, Fe^3+^ is converted back to Fe^2+^ by six-transmembrane epithelial antigen of prostate 3 (STEAP3), releasing Fe^2+^ into the cytoplasm for use in various cellular processes, including those in the cytosol and mitochondria. The iron that is not used in the cytoplasm is stored in ferritin or secreted via ferroportin 1 (FPN1). Changes in iron metabolism can directly promote ferroptosis, contributing to the development and progression of diseases such as atherosclerosis, which is characterized by lipid metabolism disorders, endothelial damage, oxidative stress, inflammation, and immune dysfunction. High iron levels have been associated with increased atherosclerosis, promoting oxidative stress and inflammatory responses linked to ferroptosis. Atherosclerosis involves the accumulation of plaques within arterial walls, leading to atherosclerotic CVDs, which can manifest as myocardial infarction, angina, stroke, peripheral arterial disease, or aortic aneurysm [16].

In vivo studies have shown that Fer-1, a ferroptosis inhibitor, can reduce atherosclerotic lesions and lipid peroxidation induced by high-fat diets [17]. During ferroptosis, the lipid peroxidation of esterified polyunsaturated fatty acids (PUFAs) is particularly pronounced due to the presence of bis-allylic hydrogen atoms compared to desaturated PUFAs, making them more susceptible to lipid peroxidation [18]. Esterified PUFAs, such as arachidonic acid (AA) and adrenic acid (AdA), are prime substrates for lipid peroxidation. During their oxidation, these PUFAs are catalyzed into acyl-CoA esters and subsequently reacylated into lysophospholipids by enzymes such as acyl-CoA synthetase long-chain family member 4 (ACSL4) and lysophosphatidylcholine acyltransferase 3 (LPCAT3). These lysophospholipids are then oxidized by LOX, leading to cell membrane rupture and ferroptotic cell death (Figure 1) [16]. The products of lipid peroxidation, including lipid hydroperoxides and aldehydes like 4-hydroxynonenal, accumulate during ferroptosis, destabilizing the cell membrane and leading to pore formation, which further promotes ferroptosis [14]. Interestingly, supplementation with PUFAs such as AA and AdA can promote ferroptosis, while monounsaturated fatty acids (MUFAs) like oleic acid can suppress it by reducing lipid peroxidation [19,20]. MUFAs displace PUFAs from plasma membrane phospholipids, thereby decreasing ferroptosis. Additionally, a high-fat diet promotes the systemic accumulation of lipids and their metabolites, further suppressing ferroptosis [20]. These findings suggest that different dietary lipid components can modulate ferroptosis in vascular smooth muscle cells [17]. The research highlights the connection between ferroptosis and CVDs, such as atherosclerosis, suggesting that targeting ferroptosis by altering dietary PUFAs to MUFAs may decrease atherosclerosis progression. In the following sections, we will continue to discuss the relationship between ferroptosis and CVDs.

## 3. Inflammation Signaling and Ferroptosis

### 3.1. Damage-Associated Molecular Patterns (DAMPs) and Ferroptosis

Inflammation is an immunological response that helps combat pathogens during host defense and facilitates tissue repair following damage [21]. This process involves cytokines and chemokines, which trigger and sustain inflammation, leading to increased production of ROS. ROS generation results in oxidative stress, lipid peroxidation, and subsequent oxidative damage. Inflammation is implicated in a range of chronic diseases, including CVDs, neurodegenerative disorders, autoimmune conditions, obesity, type 2 diabetes, endocrine disorders, osteoporosis, cancer, colitis, Crohn’s disease, and metabolic dysfunction-associated steatohepatitis (MASH) [22,23].

Interestingly, the inhibition of ferroptosis can ameliorate clinical symptoms of colitis; however, it can also promote colon tumorigenesis, highlighting a dual role of ferroptosis in intestinal diseases [24,25]. In addition, ferroptosis inhibitors have been shown to repress hepatic lipid peroxidation, thereby reducing the severity of MASH [26]. This suggests that ferroptosis plays a fundamental role in the pathophysiology of inflammation and could serve as a potential therapeutic target. Indeed, numerous experiments have confirmed the involvement of ferroptosis in inflammatory processes, with solid evidence indicating that ferroptosis can both induce and accelerate inflammation through its immunogenic effects [27]. Ferroptosis also triggers an inflammatory response by releasing damage-associated molecular patterns (DAMPs), which are immunogenic. Once released from cells, DAMPs promote a non-infectious inflammatory response by binding to pattern recognition receptors (PRRs) [27]. For instance, high mobility group box 1 (HMGB1), a DAMP released during ferroptotic cell death, binds to PRRs and drives inflammation by activating macrophages to produce pro-inflammatory cytokines [28].

During inflammation, the production of DAMPs is accompanied by increased AA production by phospholipase A2 (PLA2). AA is subsequently metabolized by cyclooxygenase-2 (COX2) into bioactive prostaglandins, which further activate macrophages and other inflammatory cells, including neutrophils and T and B lymphocytes. The inflammatory response significantly elevates ROS production, which interacts with lipids, particularly PUFAs. PUFAs play a crucial role in cellular processes related to inflammation by inducing lipid remodeling in immune cells. This lipid remodeling occurs by incorporating phospholipids, such as phosphatidylethanolamine (PE), which are converted to PUFA-PE by the enzymes ACSL4 and LPCAT3 [29]. Additionally, proteins such as aldoketoreductase family 1 member C1 (AKR1C1), ChaC glutathione-specific gamma-glutamylcyclotransferase 1 (CHAC1), ferritin heavy chain 1 (FTH1), and prostaglandin-endoperoxide synthase 2 (PTGS2) are associated with lipid metabolism, glutathione metabolism, and iron storage. Their deregulation can lead to cellular ferroptosis [30]. For instance, AKR1C1 prevents ferroptosis by converting lipid peroxidation end products into non-toxic lipid-derived alcohols [31]. CHAC1 expression is induced by cystine starvation-triggered ferroptosis [32], and FTH1 is generally upregulated during ferroptosis [33].

Ferroptotic cells can release AA, leading to the synthesis of eicosanoids and enhanced LOX activity through the massive release of oxidized lipid mediators. When AA is released from phospholipids by PLA2 and phospholipase C (PLC), it serves as a precursor for bioactive proinflammatory mediators, including prostaglandins, interleukin (IL)-1, IL-6, and tumor necrosis factor (TNF), which drive inflammatory cascades. These cytokines, along interferon-γ (IFN-γ), are involved in tissue iron storage and ferritin synthesis regulation. An abnormal inflammatory response may contribute to iron metabolism disorders and affect the redox system balance. For example, nuclear receptor coactivator 4 (NCOA4) regulates ferritinophagy, leading to the degradation of ferritin by autophagolysosomes. This process results in intracellular iron overload, triggering oxidative stress and exacerbating inflammation [34,35,36,37]. Collectively, these findings indicate that inflammation molecules, such as AA, can induce ferroptosis through various signaling pathways associated with the inflammatory response [34,35,36,37].

### 3.2. JAK-STAT Cell Signaling Pathway and Ferroptosis

Several signaling pathways and regulatory mechanisms are tightly linked to ferroptosis-related inflammation. One such pathway is the JAK-STAT cell signaling pathway. When IL-6, TNF, and IL-1β bind to their respective receptor, they activate Janus kinases (JAK) through phosphorylation. Activated JAKs then phosphorylate signal transducers and activators of transcription (STAT), leading to their dimerization and nuclear translocation. This process promotes the transcription of target genes, such as STAT3, which increases hepcidin expression. Elevated hepcidin levels inhibit iron export, contributing to ferroptosis [38,39].

### 3.3. NF-κB and Ferroptosis

The nuclear factor-kappa B (NF-κB) signaling pathway is also involved in ferroptosis-related inflammation. NF-κB plays a crucial role in chronic diseases by upregulating the transcription of pro-inflammatory genes, resulting in elevated levels of IL-1, TNF, and IL-6. These cytokines are associated with CVDs, neurodegenerative disorders, endocrine and metabolic alterations, and autoimmune conditions [23]. NF-κB is also activated by DAMPs, such as HMGB1. Released by ferroptotic cells, HMGB1 downregulates antioxidant gene transcription, thereby enhancing oxidative stress. Additionally, NF-κB influences iron metabolism during inflammation by promoting the secretion of lipocalin 2 (LCN2), which facilitates extracellular iron transport into cells. NF-κB can also be activated by ROS, a key feature of ferroptosis, generating inflammatory mediators like TNF, CXC chemokine ligand 1 (CXCL1), C-X-C motif chemokine ligand 8 (CXCL8), and colony stimulating factor 2 (CSF2) [40].

### 3.4. MAPK and Ferroptosis

The mitogen-activated protein kinase (MAPK) pathway is another cell signaling pathway influenced by intracellular iron overload and excessive lipid peroxidation. This leads to the phosphorylation of extracellular signal-regulated kinases 1 and 2 (ERK1/2) and increased phosphorylation of c-FOS and p38 MAPK, which in turn heightens oxidative stress. This stress is associated with the synthesis of IL-1β, IL-6, and IL-18, and reduced expression of solute carrier family 7 member 11 (SLC7A11) and GPX4, both critical components of anti-ferroptotic mechanisms. The MAPK pathway may mediate neuroinflammation by releasing iron from the labile iron pool during ferritinophagy, suggesting a link between MAPK and ferroptosis [41]. c-FOS and p38 MAPK are Ser/Thr kinases activated by MAPK in response to extracellular stress, infections, ischemia, DNA damage, oxifative stress, and cytokines [42,43]. These kinases phosphorylate numerous substrates, playing pivotal roles in stress adaptation, inflammation, and tumor formation. p38 MAPKs are involved in ischemia-reperfusion injury, heart failure, arrhythmias, Alzheimer’s disease, epilepsy, and tumorigenesis. Inhibiting p38 MAPK may provide a systemic anti-inflammatory effect and target diseases with inflammatory components, such as atherosclerosis [44]. Ferroptotic cell death is associated with OS due to lipid peroxidation and inflammation through various signaling pathways, including MAPK. The decreased levels of SLC7A11 and GPX4 during ferroptosis and MAPK activation highlight the involvement of the MAPK pathway in lipid peroxidation-induced oxidative stress. Further studies are needed to explore the association between ferroptosis mechanisms and p38 MAPK in related diseases.

### 3.5. cGAS-STING and Ferroptosis

Ferroptosis also activates the cyclic GMP-AMP synthase-stimulator of interferon genes (cGAS-STING) signaling pathway, which generates ROS and sensitizes cells to ferroptosis. When erastin induces mitochondrial oxidative stress, it increases mitochondrial translocation of STING [45,46]. STING is a PRR that binds cyclic dinucleotides (CDNs) produced by microorganisms or the cytoplasmic cGAS. STING can bind to host or pathogen-derived double-stranded DNA (dsDNA), including nuclear DNA (nDNA) or mitochondrial DNA (mtDNA), thereby activating the signaling pathway [47]. Ferroptosis-induced mitochondria damage could contribute to dsDNA release, further activating the cGAS-STING pathway. Increased lipid peroxidation may activate this pathway through STING carbonylation, which inhibits innate antiviral immune responses and reduces the recognition of DAMPs or pathogen-associated molecular patterns (PAMPs) in peritoneal macrophages [48]. Additionally, the cCAS-STING signaling pathway appears to be involved in the inflammasome pathway, where NLPR3 may initiate further oxidative stress, lipid peroxidation, and ferroptosis.

### 3.6. NLRP3 and Ferroptosis

Heme acts as a significant pro-oxidant and proinflammatory mediator and is classified as a classic alarmin capable of triggering the activation of the NLRP3 inflammasome in macrophages [49,50]. Additionally, endothelial cells can also respond to different alarmins by activating the NLRP3 inflammasome, leading to an increased production of IL-1β. This mechanism plays an important role in several pathological conditions, including atherosclerosis [51]. In this context, heme-derived ROS have a profound impact on endothelial cells, signaling by nitric oxide dismutase (NOD), leucine-rich repeats (LRR), and pyrin domain-containing proteins (NLRP) [52]. This process results in elevated IL-1β secretion and the exocytosis of Weibel–Palade bodies [53]. As a consequence, activated endothelial cells upregulate adhesion proteins such as vascular cell adhesion molecule-1 (VCAM-1), intercellular adhesion molecule 1, (ICAM-1), and selectins, which promote neutrophil recruitment and inflammation. Additionally, endothelial cells increase the production of antioxidant proteins such as ferritin and heme oxygenase 1 (HO-1), with the latter playing a regulatory role in inflammation by inhibiting the NF-κB pathway [54].

Heme is released during physiological or pathological hemolysis and functions as a DAMP, binding to Toll-like receptor 4 (TLR4) and activating the NF-κB pathway. Heme can also act as an autocrine signal via TNFR1, initiating a TNF-mediated necrotic process in macrophages through the TLR4/MyD88 pathway [55]. This inflammatory-mediated endothelial damage leads to tissue injury, immune cell infiltration, and endothelial dysfunction [55]. These cells signaling pathways, such as MAPKs, cGAS-STING, and NF-κB, are closely associated with the activation of ferroptosis, contributing to inflammation in several diseases.

## 4. Macrophages and Ferroptosis

Ferroptosis in macrophages is a key event in advanced atherosclerotic plaques, contributing to the formation of necrotic cores and thereby exacerbating the progression of atherosclerosis [10,11]. Elevated uric acid levels have been shown to enhance macrophage ferroptosis, accelerating atherosclerosis [12]. When macrophages take up oxidized low-density lipoproteins (ox-LDL), they release inflammatory cytokines, including IL-1, IL-3, IL-6, IL-8, IL-18, and TNF. In particular, IL-1β and TNF have atherogenic effects, promoting the expression of surface molecules such as ICAM-1, VCAM-1, CD40, and selectins on endothelial cells, smooth muscle cells, and macrophages [56].

Recent studies suggest a bidirectional relationship between macrophage polarization and ferroptosis, where they influence each other at the cellular level or through intercellular communication in a context-dependent manner [57]. For instance, M1 macrophages are resistant to ferroptosis due to the loss of arachidonate 15-lipoxygenase (ALOX15) activity [57]. In atherosclerosis, macrophages contribute to tissue repair and the proliferation of vascular smooth muscle cells, enhancing plaque stability [13]. The link between macrophages and ferroptosis underscores the significance of ferroptosis-induced inflammation in various diseases. This cell death process releases DAMPs into the extracellular space, which are recognized by macrophages through their PRRs, leading to the synthesis and release of proinflammatory mediators such as IL-1, IL-6, IL-8, and TNF. Iron overload in macrophages promotes ferroptosis and activates inflammatory responses, including increased secretion of matrix metalloproteinases, thereby advancing atherosclerosis [58].

Some studies suggest that targeting M2 macrophages to prevent ferroptotic cell death as a novel therapeutic approach for inflammatory diseases [59]. However, research by Marques et al. showed that macrophages exposed to high levels of oxLDL and proinflammatory cytokines upregulate heme oxygenase (Hmox1), H-ferritin (Fth1), hepcidin (Hamp), and *FPN1* mRNA levels. Despite this, only FPN1 failed to increase its protein levels, leading to iron retention in macrophages due to reduced export and increased storage. This iron overload could contribute to plaque instability in atherosclerosis [60].

Macrophages acquire iron from two primary sources: phagocytized red blood cells (RBC) and extracellular ferric iron (Fe^3+^). The former relies on Hmox1 activity to degrade heme and produce ferrous iron (Fe^2+^), while the latter involves the coupling of TF with its receptor (TRF1) [60]. Hepcidin, a key regulator for iron homeostasis, is upregulated by iron and inflammation [61]. Hepcidin targets and degrades FPN1, decreasing iron export and increasing iron accumulation, making macrophages more susceptible to ferroptosis. Hepcidin also enhances ox-LDL uptake and reduces cholesterol excretion in macrophages via autocrine hepcidin formation, induced by iron excess and high ox-LDL levels, thereby aggravating atherosclerosis through a vicious cycle [62].

“Efferocytosis”, a mechanism for debris clearance by macrophages, is crucial in preventing secondary necrosis and stimulating the release of anti-inflammatory cytokines such as IL-35 and IL-37 [13]. These mechanisms illustrate how ferroptosis plays a pro-inflammatory role in atherosclerosis.

## 5. Protective Role of Interleukin-37 in Cardiovascular Diseases

Interleukins are proteins critically involved in the pathogenesis and progression of inflammatory and autoimmune diseases, playing a significant role in cardiomyocyte apoptosis by modulating the balance between cell survival and death in the heart [63]. Among these, IL-37, a member of the IL-1 family, has gained attention for its role in regulating inflammation [64,65,66]. IL-37 is expressed in diverse human tissues, including the colon, skin, tonsils, esophagus, placenta, and several tumor types such as melanoma, breast, prostate, and colon cancers [67]. It exerts its anti-inflammatory effects by reducing the production of inflammatory interleukins induced by TLRs [68]. Moretti et al. [69] reported that IL-37 inhibits the activation of Th2/Th17 cells in mice with allergic aspergillosis, suggesting a potential role of IL-37 in modulating the adaptive immune response.

Further studies have shown that Smad3, a downstream mediator of the transforming growth factor-β (TGF-β) superfamily, is integral to the immunosuppressive and immunoregulatory properties of IL-37 in both in vitro and in vivo settings. Inhibiting Smad3 activation or reducing its expression has been shown to decrease the immunoregulatory activity of IL-37 [67,68,69]. Despite these findings, the specific effects of the IL-37-Smad3 signaling pathway and its subsequent impact on inflammatory mediators during the development of CVDs remain largely unexplored. There is evidence suggesting that Smad3 plays a crucial role in atherosclerosis, a chronic inflammatory condition characterized by the release of inflammatory interleukins. Consequently, IL-37 emerges as a promising immunoregulatory cytokine with significant potential in the context of atherosclerosis, particularly as an inflammation inhibitor that can shift the cytokine balance away from excessive inflammation.

Regulatory T cells (Tregs) are key mediators of the immunoregulatory response and provide protection against atherosclerosis. Enhancing anti-inflammatory responses may reduce the levels of pro-inflammatory mediators, making this approach a potential target for therapeutic development. Tregs secrete regulatory interleukins, primarily IL-10, TGF-β, and IL-35 [70].

Lotfy et al. [71] reported that IL-37 levels were elevated in patients with chronic lower limb atherosclerotic ischemia compared to non-atherosclerotic controls, and the circulating levels of IL-37 correlated with disease severity. Interestingly, supplementation with recombinant IL-37 (rIL-37) increased the release of IL-10 and TGF-β in supernatants of T cells co-cultured with Tregs, suggesting that IL-37 may modulate suppressive functions in the atherosclerotic process, potentially enhancing Treg inhibitory function with a marked response in severe disease [71]. Although previous studies have indicated a possible association between IL-37 and acute coronary syndrome (ACS), the specific relationship and its predictive value for patient prognosis remain to be clarified.

Ji et al. demonstrated a significant reduction in certain regulatory interleukins in patients with ACS [72]. Furthermore, their study indicated that IL-37 is synthesized in foam-like cells within atherosclerotic coronary and carotid artery plaques and that its levels negatively correlate with left ventricular ejection fraction in ACS patients [71]. However, the exact role of altered IL-37 levels in the progression of atherosclerosis and the onset of ACS is still not fully understood.

Further mechanistic studies, such as those involving the administration of exogenous IL-37 (e.g., recombinant human IL-37 or IL-37 transgenic mice), are necessary to elucidate the precise role of IL-37 in atherosclerosis. Chai et al. [73] suggested that IL-37 has a significant role in atherosclerosis, demonstrating in apoE-deficient diabetic mice that IL-37 treatment significantly reduced vascular calcification and atherosclerosis progression. IL-37-treated mice exhibited reduced calcification areas, as detected by von Kossa and alizarin red staining, and decreased expression of bone morphogenetic protein-2 (BMP-2), TNF, IL-18, and IL-10 in atherosclerotic lesions. This group also observed smaller plaque sizes and lower plaque vulnerability scores in the aortic root. The upregulation of IL-10 expression and concomitant reduction in TNF and IL-18 production were identified as direct immunoregulatory mechanisms through which IL-37 ameliorated atherosclerosis and coronary artery calcification [73,74].

In patients with atherosclerosis, higher IL-37 levels have been detected in calcified samples, particularly within macrophages and vascular smooth muscle cells. Additionally, patients with coronary artery calcification showed significantly increased plasma IL-37 concentrations, with correlation analysis revealing positive associations between IL-37 levels and age, fasting glucose, alkaline phosphatase, IL-6, TNF, and C-reactive protein [75].

Interestingly, IL-37 is also involved in regulating cholesterol homeostasis, reducing plasma cholesterol, fatty acids, and triglycerides levels [76]. Recent studies have identified IL-37 polymorphisms, such as rs2708961, rs2723187, and rs2708947, which are associated with a decreased risk of hypercholesterolemia in the Mexican population. These polymorphisms have also been linked to cardiovascular risk factors. Some IL-37 polymorphisms have been associated with cardiometabolic factors in individuals with and without hypercholesterolemia [76]. Additionally, Yin et al. reported an association between the IL-37 rs3811047 polymorphism and coronary artery disease, along with decreased IL-37 mRNA expression levels [77]. Xu et al. [78] also observed that IL-37 mitigates the toxic effects of inflammatory mediators on myocardial cells and ameliorates myocardial infarction. Wu et al. [79] demonstrated that IL-37 protects mouse cardiomyocytes from apoptosis under ischemia/reperfusion conditions and suppresses the production of inflammatory mediators, including cytokines, chemokines, and neutrophil infiltration. This protective effect contributes to reduced cardiomyocyte apoptosis and ROS generation, and the immunoregulatory function of IL-37 was shown to inhibit TLR-4 expression and NF-κB activation after ischemia/reperfusion, while increasing regulatory IL-10 concentrations [79].

In a recent study, Law et al. [80] reviewed the clinical implications of IL-37 in cardiovascular manifestations of COVID-19. Elevated circulating IL-37 levels were found in COVID-19 patients, with higher IL-37 levels being associated with shorter hospitalization periods. This suggests that IL-37 may provide protective effects during COVID-19 infection, further highlighting its potential protective role in CVD.

## 6. Interleukin-37 and Ferroptosis in Cardiovascular Diseases

IL-37 has been implicated in mitigating macrophage ferroptosis, a regulated form of cell death associated with lipid peroxidation, which has been increasingly recognized as a contributor to CVDs. Recent studies have explored the relationship between ferroptosis and the protective effects of IL-37 in macrophages exposed to high glucose and ox-LDL, particularly in diabetic ApoE−/− mice. Xu et al. found that IL-37 treatment significantly reduced plaque area, improved lipid profiles, decreased serum levels of inflammatory mediators such as IL-1β and IL-18, and upregulated the expression of GPX4 and Nrf2 in the aortas of these diabetic mice [17].

In vitro experiments further demonstrated that IL-37 inhibited high glucose/ox-LDL-induced ferroptosis in macrophages. This was evidenced by improved cell membrane integrity, reduced malondialdehyde production, and increased GPX4 expression. Additionally, the study revealed that IL-37 promoted the activation of the Nrf2 pathway, which is known to inhibit ferroptosis in macrophages (Figure 2) [10]. Nrf2, a transcription factor, plays a critical role in maintaining cellular homeostasis against oxidative stress by regulating the expression of antioxidant enzymes. Its activation leads to the upregulation of anti-ferroptotic genes, such as GPX4, SLC7A11, and *FPN1*, and the downregulation of *ACSL4*. Numerous studies have highlighted the beneficial effects of the NRF2 pathway in alleviating ferroptosis across various conditions, including myocardial infarction, colitis, ischemia-reperfusion injury, and ischemic stroke [9,80,81,82]. However, whether IL-37 directly or indirectly facilitates the dissociation of Nrf2 from its inhibitor, Kelch-like ECH-associated protein 1 (KEAP1), remains unknown and warrants further investigation.

While the role of IL-37 in atherosclerosis is well established, its involvement in macrophage ferroptosis and its potential as a therapeutic target in cardiovascular diseases represent emerging areas of interest. As the mechanisms underlying ferroptosis become more clearly defined, targeting ferroptosis could offer significant therapeutic benefits for several cardiovascular conditions.

## 7. Conclusions and Closing Remarks

Inflammation is a key driver in the production of ROS and oxidative stress, which ultimately leads to ferroptosis, a distinct form of regulated cell death. Our review highlights the critical role of ferroptosis in the pathophysiology of various diseases, including CVDs, where it contributes to disease progression through its unique mechanistic and morphological features. Understanding the regulatory pathways that govern ferroptosis opens new avenues for exploring disease mechanisms and identifying novel therapeutic targets.

IL-37 emerges as a promising candidate in this context, given its potential to regulate ferroptosis through its anti-inflammatory properties. By reducing inflammation, IL-37 could serve as an effective intervention strategy for preventing and treating CVDs, where inflammation-driven ferroptosis plays a detrimental role. Our discussion underscores the high potential of IL-37 in mitigating ferroptosis in cardiovascular conditions, offering a novel therapeutic approach that targets the inflammatory pathways involved in this form of cell death.

## Figures and Tables

**Figure 1 ijms-25-09758-f001:**
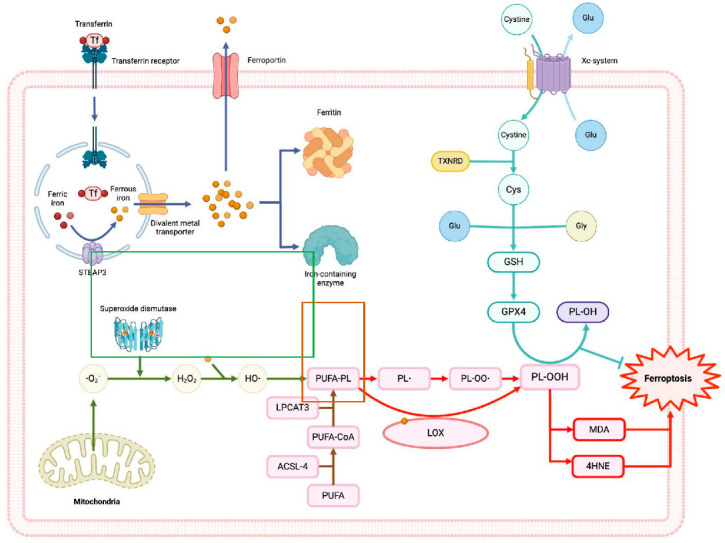
**Ferroptosis mechanisms and the system Xc^−^-GSH-GXP4 pathway**. Iron metabolism (blue lines): The transferrin receptor (TRF1) mediates the endocytosis of the ferric iron (Fe^3+^)-transferrin complex, then the ferric iron is reduced to ferrous iron (Fe^++^) by the metalloreductase STEAP3, leading to the entrance to the cytosol by the divalent metal transporter (DMT1). In the cytosol, iron can form the labile iron pool (LIP), couple with other enzymes, be stored by ferritin, or be exported by ferroportin (FPN). Lipid metabolism (brown lines): The enzymes ACSL4 and LPCAT3/5 catalyze the formation of membrane phospholipid-containing polyunsaturated fatty acid chains (PUFA-PL). Reactive oxygen species (green lines): Superoxide anion (^●^O_2_^−^) is produced in mitochondria and then is reduced to peroxide hydrogen (H_2_O_2_) by superoxide dismutase (SOD). Lipid peroxidation (red lines): The non-enzymatic pathway initiates with the production of hydroxyl radicals (^●^OH) via Fenton/Haber–Weiss reactions, a ferrous iron-catalyzed H_2_O_2_ oxidation process. The ^●^OH react with PUFAs, which leads to lipid radicals and propagates lipid peroxidation. Lipoxygenase (LOX) is essential for the enzymatic pathway by catalyzing PUFAs into lipid hydroperoxide (PL-OOH). Malondialdehyde (MDA) and 4-hydroxynenanol (4HNE) are the final metabolites of lipid peroxidation. System Xc^−^–GSH–GPX4 pathway (cyan lines): System Xc^−^regulates cystine importation and glutamate exportation; cystine is reduced to cysteine by thioredoxin reductase (TXNRD) so it can be used to form glutathione (GSH). The enzyme glutathione peroxidase 4 (GPX4) utilizes GSH to reduce PL-OOH into lipid alcohols (PL-OH) and inhibit lipid peroxidation.

**Figure 2 ijms-25-09758-f002:**
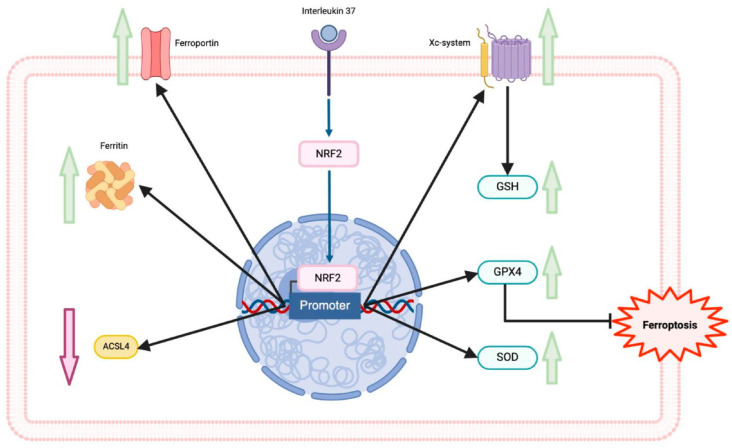
Interleukin 37 (IL-37) and nuclear factor erythroid 2-related factor 2 (Nrf2) pathway. The master regulator of antioxidant response Nrf2 is activated under significant oxidative stress, and IL-37, then it is translocated to induce its target genes in the nucleus. The Nrf2 pathway upregulates proteins of the iron metabolism, superoxide dismutase, and the system Xc^−^–GSH–GPX4 pathway and downregulates ACSL4. IL-37 can induce the activation and translocation of Nrf2 to the nucleus. Glutathione (GSH), glutathione peroxidase 4 (GPX4), acyl-CoA synthetase long chain family member 4 (ACSL4).

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
