# Peer review of "The Protective Role of Interleukin-37 in Cardiovascular Diseases through Ferroptosis Modulation"

_ijms, 2024, doi:10.3390/ijms25189758_

Round 1

Reviewer 1 Report

Comments and Suggestions for Authors

Authors discussed the potential Protective Role of interleukin-37 via Ferroptosis in Cardiovascular Diseases. the topic is of interest but the manuscript needs to be rewritten, taking in consideration these comments.

Section 2: no discussion between Ferroptosis and CVDs interaction.

Section 3: so long . better to shorten or divide on subsection , better for reader  

-          Paragraph 7 from line 246 to the end, there is no discussion on link with ferroptosis

Section 4: line 309 to 314 better to add interaction with ferroptosis

Minor

-          line 232. add ref after “pathways…)

Comments on the Quality of English Language

 Minor editing of English language required.

Reviewer 2 Report

Comments and Suggestions for Authors

This was an interesting review of ferroptosis and the potential role of IL-37 in preventing ferroptotic cell death in cardiovascular disease.  The figures were clear and the figure legends very helpful for the reader.

The  review itself would benefit from focusing more on IL-37, ferroptosis particularly in CVD  instead of presenting a very in depth review of ferroptosis in a wide variety of tissues.

Minor edits.

1.  Line 71- change no enzymatic to non-enzymatc

2.  Line 133-  what is meant by six membrane epithelial cells?  

3. line 135- just a look should be changed to this a brief overview ( More formal writing and less slangy)

4. Are any ferroptotic inhibitors used clinically?  Used in CVD? If so add to the text.

5. line 199,   spello out ACSL4 and LPCAT3  

6.Page 12,  there is alteration between the use of hepcidin and hamp.  I would suggest sticking with hepcidin to be clear.

7.  Line 359-   "main protein for receptors" to "downstream mediators" or "direct targets"

8,  362 space is needed between Smad3 and knockdown

9. 377 TGFb has been defined you do not need to define again.

10.  Line 398 was unclear to me.  Please rewrite.

Comments on the Quality of English Language

English was fine

Author Response

Please see the attchment

Round 2

Reviewer 1 Report

Comments and Suggestions for Authors

Authors respond adequately to previous comments and make changes accordingly. Interaction between Ferroptosis and CVDs may be discussed more. The paper is acceptable for publication. 

Comments on the Quality of English Language

Minor revision is required before final submission